# Are routinely collected clinical and sociodemographic characteristics associated with social functioning and activities of daily living in schizophrenia? A machine learning approach descriptive of a schizophrenia cohort

Guillaume Barbalat[1]*, Julien Plasse[1], Isabelle Chéreau-Boudet[2], Benjamin Gouache[3], Nathalie Guillard-Bouhet[4], Emilie Legros-Lafarge[5], Nicolas Franck[1]

1 Centre Ressource de Réhabilitation Psychosociale et de Remédiation Cognitive (CRR), Hôpital Le Vinatier, Centre National de la Recherche Scientifique (CNRS UMR 5229) et Université Lyon 1, Lyon, France, 2 Centre Référent Conjoint de Réhabilitation (CRCR), Centre Hospitalier Universitaire de Clermont-Ferrand, Clermont-Ferrand, France, 3 Centre Référent de Réhabilitation Psychosociale et de Remédiation Cognitive (C3R), Centre Hospitalier Alpes Isère, Grenoble, France, 4 Centre de REhabilitation d'Activités Thérapeutiques Intersectoriel de la Vienne (CREATIV), Centre Hospitalier Laborit, Poitiers, France, 5 Centre Référent de Réhabilitation Psychosociale de Limoges (C2RL), Limoges, France

* Guillaume.Barbalat@ch-le-vinatier.fr

## Abstract

Patients with schizophrenia often experience substantial impairments in social functioning and activities of daily living (ADLs). Previous studies have highlighted links between sociodemographic and clinical factors and daily life functioning. However, routinely collected, non-scale-based clinical and sociodemographic characteristics have rarely been systematically evaluated as a standalone predictive framework. Understanding whether such variables alone can inform functional outcomes in schizophrenia may have important implications for public health. Using the French multicentric psychosocial rehabilitation database REHABase, we predicted five dimensions of the Social Autonomy Scale (a validated, clinician-administered questionnaire widely used in France): personal care, basic ADLs, financial autonomy, complex ADLs, and social and affective relationships. We used a SuperLearner ensemble machine learning method with a large set of routinely collected sociodemographic and basic clinical predictors descriptive of schizophrenia patients. Our sample comprised 948 participants. Averaged R2 on hold-out testing sets were higher for basic ADLs (mean R2: 0.35) than for social and affective relationships (0.16), financial autonomy (0.14), complex ADLs (0.13), and personal care (0.01). Factors associated with improved functioning included: being in a relationship, higher education, lower Clinical Global Impression scores, higher Global Assessment of Functioning scores, living in personal housing, being employed, mid-range illness duration, a history of suicide attempts and psychiatric comorbidities. Our findings

**Data availability statement:** There are ethical and legal restrictions on sharing our de-identified data set as data contain potentially identifying or sensitive patient information. Data requests may be sent to our internal committee ("Comite d'Acces aux Donnees de REHABase", CAD-R) by emailing 1879@ch-le-vinatier.fr. To ensure the reproducibility of our predictive models and transparency of our methodology, the R code utilized for data preprocessing, cross-validation, and performance evaluation is publicly available at: https://github.com/gbar-balat/SocioDemoClin-Rehab/. This repository contains all scripts and necessary documentation required to replicate the reported statistical analyses.

**Funding:** The author(s) received no specific funding for this work.

**Competing interests:** The authors have declared that no competing interests exist.

indicate an association between socio-demographic and standard clinical variables routinely assessed in practice and outcomes in social functioning and ADLs. However, these variables account for only a limited proportion of the observed variance. This underscores the need for more specialized and precise assessments, e.g., based on cognitive abilities, to better understand and address patient functioning. We recommend targeted interventions focused on improving clinical symptoms, housing conditions, and supporting employment. Finally, clinicians should not assume that patients with seemingly protective factors, such as shorter illness duration, or absence of comorbidities, do not require further support.

## Introduction

Patients with schizophrenia are characterized by impairments in their social lives, including relationships with family, friends, and intimate partners, as well as in activities of daily living (ADLs) such as shopping, cooking, cleaning, using transportation, and managing finances. Functioning impairments are directly responsible for much of the high societal costs associated with schizophrenia. In the United States for example, the costs related to unemployment, lost productivity, and caregiving for individuals with schizophrenia are estimated at $174 billion, accounting for approximately half of the total economic burden of the disease [1]. In addition, impaired functioning in schizophrenia has been associated with an increased risk of suicide attempts [2], as well as with poorer quality of life and recovery outcomes [3,4]. Functional deterioration can also contribute to the worsening of clinical symptoms themselves [5–7].

Contemporary neuroscience models the neural dysfunctions associated with schizophrenia, translating them into quantitative biomarkers and behavioral phenotypes. Although these neural signatures are pivotal for elucidating the underlying pathophysiology of the condition, they remain absent from routine clinical assessment. From a clinical standpoint, research has shown that impairments in social functioning and ADLs among patients with schizophrenia are primarily related to several key factors, among which negative symptoms, such as loss of volition, withdrawal, inactivity, and muteness; and cognitive deficits in non-social (e.g., working memory, attention, executive functions) and social cognitive domains (e.g., emotional processing and mentalizing) [8]. Socio-demographic and illness-related factors also play a significant role. Factors most commonly associated with the intensity or course of psychosocial disabilities include treatment modalities (inc. medication compliance [9], and side effects [10], psychopathological symptoms, and socio-demographic variables such as employment status, gender, marital status, education, age, and accommodation type [11]. Other factors associated with varying levels of ADL performance include country [9] and type of residence [12], duration of illness [13–15], number of admissions [16], physical [16] and addiction comorbidities [13].

Measures such as symptom-specific scales (e.g., the Positive and Negative Syndrome Scale, PANSS) are not always available in routine clinical practice, where assessments are often limited to demographic information, illness history, treatment

variables, and global measures such as the Clinical Global Impression (CGI) or the Global Assessment of Functioning (GAF). If these routinely collected, non-symptom-based variables are strongly associated with social functioning and ADLs, they could serve two important purposes. First, validating the predictive value of routinely collected measures would help ensure that clinicians use the most effective tools for estimating patient functioning. These measures could aid in identifying individuals at higher risk for poor social functioning or difficulties with daily activities, thereby enabling earlier and more targeted interventions. Such an approach may complement – and in some cases precede – more specialized assessments based on symptoms and cognition, offering greater applicability in real-world clinical settings. Conversely, if these measures prove insufficient, symptom- or cognition-based assessments should be prioritized to provide a more accurate evaluation and better patient support. Second, identifying which specific variables – and which thresholds within them – are most predictive could clarify how these factors contribute to real-life challenges. This knowledge could also inform public health strategies aimed at improving social functioning and daily life outcomes for patients with schizophrenia.

Despite its potential significance, to our knowledge, existing literature lacks a comprehensive analysis examining how routinely collected socio-demographic and clinical factors predict social functioning and ADLs in individuals with schizophrenia as a standalone predictive framework. An optimal study in this area would employ a model that (1) incorporates a wide range of variables routinely collected; (2) tests various types of effects, including linear, non-linear, and interaction effects [17]; and (3) is evaluated rigorously (i.e., using an independent dataset). Using such an analytical framework, we examined whether socio-demographic and clinical factors routinely collected are effectively associated with social functioning and ADLs in patients with schizophrenia receiving rehabilitation services.

## Methods

### Data

**Description of the database.** We used data from the French multicentric psychosocial rehabilitation database REHABase [18], and specifically background socio-demographic data, basic clinical data, and rehabilitation outcome data from patients diagnosed with schizophrenia. Typically, socio-demographic and clinical data (the predictors analyzed in this study) are collected at service entry by a clinical psychiatrist, whereas rehabilitation outcome data (including functioning outcomes as measured in this study) are collected in the subsequent days/weeks by a psychiatric nurse. The initial dataset comprised information from 33 clinical centers. To ensure statistical robustness, we consolidated centers with fewer than 50 observations into a single group, resulting in a final set of six analytical units, comprising five individual centers and one grouped category. Rather than relying on a predetermined sample size calculation, we employed an exhaustive inclusion strategy, utilizing all available observations from the database. This approach was chosen to maximize the statistical power of the analysis and to capture the full range of clinical variability present within the study population.

**Dependent variables (functioning outcomes).** We used the social autonomy scale (SAS, "Echelle d'Autonomie Sociale"), which is a clinician-report questionnaire validated in French speaking population [19].

The scale is composed of 17 items assessing social and daily life functioning, each scored on a 7-point Likert scale (0: no problem; 6: major issues), yielding a total score range of 0–102 – lower scores indicate better functioning. Items are scored by the case manager following discussions with both the patient and their family members. In the validation study, the Cronbach's alpha was 0.86, indicating high internal consistency, and the inter-rater reliability was 0.936, reflecting excellent agreement between raters [19].

Items of the SAS are grouped into 5 dimensions. Personal care (PER) consists of 3 items evaluating body hygiene, hygiene of personal belongings, and eating habits (score range: 0–18). Basic activities of daily living (BAS) is composed of 4 items evaluating whether the patient lives independently, their ability to maintain their accommodation, their rational use of transportation, and their capacity to run errands (score range: 0–24). Financial autonomy (FIN) is composed of 3

items evaluating the ability to manage a personal budget, build and manage assets, and earn income (score range: 0–18). Complex activities of daily living (COM) includes 4 items evaluating the ability to navigate different places, organize trips and leisure activities outside the home, appropriately use communication and information tools, and effectively organize daily activities (score range: 0–24). Finally, the social and affective functioning dimension (SOC) comprises 3 items evaluating family life, friendships, and intimate relationships (score range: 0–18).

**Predictors (socio-demographic and clinical data).** The following background clinical and socio-demographic factors, as reported by clinical psychiatrists, were identified as potential predictors of functioning outcomes: center (six categories); age (continuous); sex (male vs. female); education (no high school diploma; high school diploma; Bachelor's degree; Master's degree); marital status (single; divorced/widowed; in a relationship); being a parent (yes vs. no); housing status (homeless; group home; family home; personal residence); employment (employed (regular); employed (specialized); unemployed); being a disabled worker beneficiary (yes vs. no); being of no fixed abode (currently; in the past; no); duration of illness (less than two years; two to five years; five to 10 years; more than 10 years); number of psychiatric admissions (nil; one; two; three; four; five to 10; more than 10); total time spent in hospital (nil; three months or less; three to six months; six to 12 months; more than 12 months); psychiatric comorbidities (yes vs. no); physical comorbidities (yes vs. no); addiction comorbidities (nil; behavioral only; substance only; both substance and behavioral); history of suicide attempts (no previous attempt; one; two; three; four or more); forensic history (yes vs. no); antipsychotic medication (nil; first-generation antipsychotics; second-generation antipsychotics; both); medication for physical disorders (yes vs. no); referrer (clinician from the public healthcare system; clinician from the private healthcare system; social worker; self-referral; other); severity score at the CGI (continuous); score at the GAF (continuous).

All procedures performed in studies involving human participants were in accordance with the ethical standards of the institutional and/or national research committee and with the 1964 Helsinki Declaration and its later amendments or comparable ethical standards. The database obtained the authorizations required under French legislation (French National 429 Advisory Committee for the Treatment of Information in Health Research, 16.060bis; French 430 National Computing and Freedom Committee, DR-2017–268).

Informed consent was obtained from all individual participants included in the study.

## Analysis

Data preprocessing and predictive modeling were performed in R (version 4.3.3).

**Strategy for handling missing data.** For each outcome measure, we removed observations with missing outcome values. However, some missing values remained in the predictors set. Overall, the percentage of missing data was of 2.8%. For individual variables, missing rates varied from 0% (for center, age, sex, psychiatric comorbidities) up to 14% for CGI and GAF, and 15% for total time spent in hospital. Our general strategy aimed to retain as many observations and variables as possible and impute missing values, ensuring the maintenance of a large number of observations and reducing the risk of selection bias. However, imputing missing data for "number of psychiatric admissions" and "total duration of admission" resulted in relatively unstable parameter estimates when applying standard general linear models during preliminary analyses (assessed by a fraction of missing information (FMI) > 0.40 [20]). After having excluded both these variables, parameter estimation did not fluctuate as much (FMI < 0.10 except for CGI where FMI = 0.19). We generated m = 20 imputed datasets. Imputation was performed prior to predictive modeling and was conducted separately on the training and hold-out testing sets to prevent data leakage (see below). The imputation model incorporated the features described in the "Predictors" section, with the exception of "number of psychiatric admissions" and "total duration of admission". We employed multiple imputation by chained equations (and the R package *mice*), using predictive mean matching for numeric variables, logistic regression for binary variables, and polytomous regression for unordered categorical variables. We derived R2 goodness-of-fit statistics and SHAP feature importance values (see

below) independently for each imputed dataset. We then aggregated these results by calculating the mean and standard deviation across the 20 imputations.

**Predictive modeling.** We analyzed each of the five dimensions of the SAS separately, performing five analyses on five datasets. We employed a SuperLearner ensemble model using the *SuperLearner* R package to optimize predictions of each outcome [21]. Base learners included generalized linear models with main terms only, regularized regression, random forests, and extreme gradient boosting – with the latter two methods inherently modeling non-linearities and interactions. In addition to training each basis learner on the full set of predictors, variables pre-selection was performed using random forest variable importance (to select the top 10 predictors by mean decrease in impurity; implemented with 1000 trees; mtry = 7; nodesize = 1). We also included a general linear model with first-order interactions, using variables pre-selected by the above random forest algorithm. For each base learner, we used an "adaptive" hyperparameter tuning strategy [22] aiming to minimize logarithmic loss. Rather than an exhaustive, fixed grid search, this method dynamically manages the number of resampling iterations allocated to each hyperparameter combination. Briefly, the algorithm iteratively eliminates poorly performing parameter sets early in the process by employing an early-stopping mechanism based on model performance variance. This ensures that only the most robust parameter sets are evaluated across the full suite of resamples, significantly enhancing computational efficiency.

For each outcome, data were split into training (70%) and hold-out testing (30%) sets. We did not stratify the train-test split by center to avoid potential limitations in sample size for smaller centers (Instead, "center" was included as a categorical feature in our model to ensure the model could learn site-specific effects). Base learners and the ensemble model were trained on the training set using 8-fold cross-validation (of note, using a greater number of folds led to computational issues). We trained the model on 21 features, consistent with the list provided in the *Predictors* section, excluding "number of psychiatric admissions" and "total duration of admission" due to their high FMI (see *Strategy for handling missing data*). Model performance was assessed on both the training set and the hold-out testing set for each of the 20 imputed datasets using the coefficient of determination ($R^2$ metric).

Finally, we computed SHAP (SHapley Additive exPlanations) values for every feature and training observation across all 20 imputed datasets using the *fastshap* R package [23,24]. For a given observation, SHAP values quantify the magnitude and direction (positive/negative) of each feature's additive contribution to the model's predicted outcome. The magnitude reflects the relative importance of each feature's contribution. By design, the sum of all SHAP values for an observation, combined with the model's base value (i.e., the average prediction), equals the model's output for that observation – i.e., the predicted outcome value.

## Results

### Description of the population

Our sample comprised 948 participants. Concordant with our previous studies [18,25], approximately three-quarters of the sample were male, with a mean age of 33 years (SD = 10; **Table 1**). Males had a mean age of 32.8 years (SD = 9.7), while females had a mean age of 34.0 years (SD = 11.0). Just over 10% had children, and the mean CGI severity score was 4.1 (SD = 1.1). The majority of the cohort was prescribed second-generation antipsychotics, accounting for 600 individuals (63%). 72 patients (7.6%) were on first-generation antipsychotics alone, while patients prescribed both first- and second-generation represented 231 individuals (24%). A small minority (30 patients, 3.2%) were on no antipsychotic medication. The distribution of illness duration skewed toward chronic cases, with 414 individuals (44%) reporting a duration of 10 years or more. Patients with an illness duration between 5–10 years comprised 196 individuals (21%), followed by 153 patients (16%) in the 2–5 year range, and 125 patients (13%) with less than 2 years of illness. Regarding psychological comorbidities, the majority of the sample (693 patients, 73%) reported no such conditions, while 255 patients (27%) reported their presence. Physical comorbidity was present in 285 individuals (30%), with the remaining 657 (69%) reporting no physical comorbidities. 652 patients (69%) had no recorded suicide attempts, 143 individuals (15%) had one attempt, 54

**Table 1. Clinical and socio-demographic characteristics of the participants.**

| Characteristic | N (%)/ Mean, (SD) |
|---|---|
| **SAS: PER** | 3.8 (3.4) |
| **SAS: BAS** | 6.7 (5.2) |
| **SAS: FIN** | 9.7 (4.5) |
| **SAS: COM** | 5.7 (4.5) |
| **SAS: SOC** | 8 (4.2) |
| **AGE** | 33, (10) |
| **SEX** | |
| **Female** | 229 (24%) |
| **Male** | 719 (76%) |
| **EDUCATION** | |
| **No dipl.** | 429 (45%) |
| **HS dipl.** | 281 (30%) |
| **Bach.** | 160 (17%) |
| **Mast.** | 69 (7.3%) |
| **(Missing)** | 9 (0.9%) |
| **MARITAL STATUS** | |
| **Single** | 772 (81%) |
| **Div./Wid.** | 48 (5.1%) |
| **In a rel.** | 114 (12%) |
| **(Missing)** | 14 (1.5%) |
| **CHILDREN** | |
| **No** | 818 (86%) |
| **Yes** | 117 (12%) |
| **(Missing)** | 13 (1.4%) |
| **HOUSING** | |
| **Gp H.** | 89 (9.4%) |
| **Fam. H.** | 388 (41%) |
| **Pers. H.** | 434 (46%) |
| **Hless** | 23 (2.4%) |
| **(Missing)** | 14 (1.5%) |
| **NFA** | |
| **No** | 803 (85%) |
| **Past** | 94 (9.9%) |
| **Current** | 29 (3.1%) |
| **(Missing)** | 22 (2.3%) |
| **EMPLOYMENT** | |
| **Empl.(reg.)** | 49 (5.2%) |
| **Empl.(spec.)** | 20 (2.1%) |
| **Unempl.** | 864 (91%) |
| **(Missing)** | 15 (1.6%) |
| **DIS. WORK. BENEF.** | |
| **No** | 508 (54%) |
| **Yes** | 405 (43%) |
| **(Missing)** | 35 (3.7%) |
| **ANTIPSYCHOTICS** | |
| **Nil** | 30 (3.2%) |

*(Continued)*

**Table 1.** (Continued)

| Characteristic | N (%)/ Mean, (SD) |
|---|---|
| FGA | 72 (7.6%) |
| SGA | 600 (63%) |
| Both | 231 (24%) |
| (Missing) | 15 (1.6%) |
| **DUR. ILLNESS** | |
| <2 yrs | 125 (13%) |
| 2-5 yrs | 153 (16%) |
| 5-10 yrs | 196 (21%) |
| 10 yrs+ | 414 (44%) |
| (Missing) | 60 (6.3%) |
| **PSYCH. COMORB.** | |
| No | 693 (73%) |
| Yes | 255 (27%) |
| **SUICIDE ATT.** | |
| 0 | 652 (69%) |
| 1 | 143 (15%) |
| 2 | 54 (5.7%) |
| 3 | 24 (2.5%) |
| 4+ | 38 (4.0%) |
| (Missing) | 37 (3.9%) |
| **ADDICTIONS** | |
| Nil | 385 (41%) |
| Behav. | 26 (2.7%) |
| Subst. | 486 (51%) |
| Both | 34 (3.6%) |
| (Missing) | 17 (1.8%) |
| **PHYS. COMORB.** | |
| No | 657 (69%) |
| Yes | 285 (30%) |
| (Missing) | 6 (0.6%) |
| **PHYS. Rx** | |
| No | 759 (80%) |
| Yes | 177 (19%) |
| (Missing) | 12 (1.3%) |
| **Forensic Hx.** | |
| No | 774 (82%) |
| Yes | 145 (15%) |
| (Missing) | 29 (3.1%) |
| **REFERRER** | |
| Pub. HC | 704 (74%) |
| Pr. HC | 127 (13%) |
| Pat. | 47 (5.0%) |
| Soc. W. | 20 (2.1%) |
| Other | 33 (3.5%) |
| (Missing) | 17 (1.8%) |

*(Continued)*

**Table 1.** (Continued)

| Characteristic | N (%)/ Mean, (SD) |
|---|---|
| **CGI** | 4.12, (1.10) |
| (Missing) | 136 (14%) |
| **GAF** | 58, (14) |
| (Missing) | 131 (14%) |

Legend. SAS, Social Autonomy Scale; dimensions: PER, personal care; BAS, basic activities of daily living; FIN, management of financial resources; COM, complex activities of daily living; SOC, social and affective relationships; dipl., diploma; HS, High-School; Bach., Bachelor's degree; Mast., Master's degree; rel., relationship; Div./Wid., Divorced/Widowed; Gp H., Group Home; Fam. H., Family Home; Pers. H., Personal Home; Hless, Homeless; Empl., Employed; reg., regular; spec., specialized; Unempl., Unemployed; Dis Work. Benef., Disability Worker Beneficiary; FGA, first generation antipsychotic; SGA, second generation antipsychotic; Dur. Illness, Duration of illness; < 2 yrs, less than 2 years;10 yrs + , 10 years or more; Psych., Psychiatric; Comorb., Comorbidities; Suicide att., Suicide attempts; Phys., Physical; 4 + , 4 or more; Behav., Behavioral; Subst., Substance; Rx, Treatment; Hx, History; Pub. HC, Public HealthCare; Pr. HC, Private HealthCare; Soc. W., Social Worker; Pat., Patient; GAF, Global Assessment of Functioning; CGI, Clinical Global Impression.

SD, standard deviation.

(5.7%) had two, 24 (2.5%) had three, and 38 patients (4.0%) had four or more attempts. Additionally, a forensic history was absent in 774 patients (82%) and present in 145 patients (15%). Addiction patterns indicated that 385 individuals (41%) had no addictions, while 486 (51%) presented with substance-related addictions. Behavioral addictions were noted in 26 patients (2.7%), and 34 patients (3.6%) reported both substance and behavioral addictions. Finally, the majority of patients (759, 80%) were not currently receiving physical prescriptions, while 177 individuals (19%) were. Unemployment was prevalent, affecting over 90% of the sample, while just over 10% were in a relationship. Slightly more than 40% had disability worker beneficiary status, 45% had no diploma, and just over 45% were living in personal housing.

The mean total score on the SAS was 34 (SD = 16.4), which is substantially lower than the mean score reported in the original validation study, likely due to the fact that its sample consisted primarily of hospitalized individuals [19]. Subscale scores indicated that participants experienced greater difficulties in social and affective relationships and financial autonomy, while personal care was less affected (Table 1).

## Predictive performance

S1 Table shows the performance metrics across 20 imputed datasets for each of the five dimensions of the SAS. Averaged R2 for the SuperLearner ensemble on the hold-out testing sets was higher for basic ADLs (mean R2: 0.35, SD: 0.01) than for financial resources (0.14, 0.01), complex ADLs (0.13, 0.01), and social and affective relationships (0.16, 0.01). The predictive performance for personal care was very low (0.01, 0.00).

The cross-validated risks on the training set varied across the base learners, with the random forest algorithm consistently demonstrating the highest performance (mean R2 range: 0.69–0.78). However, the substantially lower $R^2$ values observed for the ensemble on the test set suggest overfitting.

S2 Table presents the contribution of each base learner to the SuperLearner ensemble across the imputed datasets. There was considerable variation in the influence of individual base learners within the ensemble.

We carried on investigating the influence of predictors for those dimensions where our model proved meaningful (i.e., not for the personal care dimension).

## Influence of predictors

S1 Fig presents the mean absolute SHAP value for each feature across the four dimensions of functioning with a meaningful R2, providing an overview of feature importance. The most influential predictors were largely consistent across

dimensions: housing, CGI, center, addictions, and education for basic ADLs; employment, education, GAF, center, and housing for financial autonomy; CGI, education, center, GAF, and housing for complex ADLs; and marital status, CGI, age, center, and housing for social and affective relationships.

Figs 1–4 display the SHAP value contributions for the ten most important predictors (excluding center) across the four dimensions. Predictors of better basic ADLs included: living in personal housing, lower CGI scores, substance addiction,

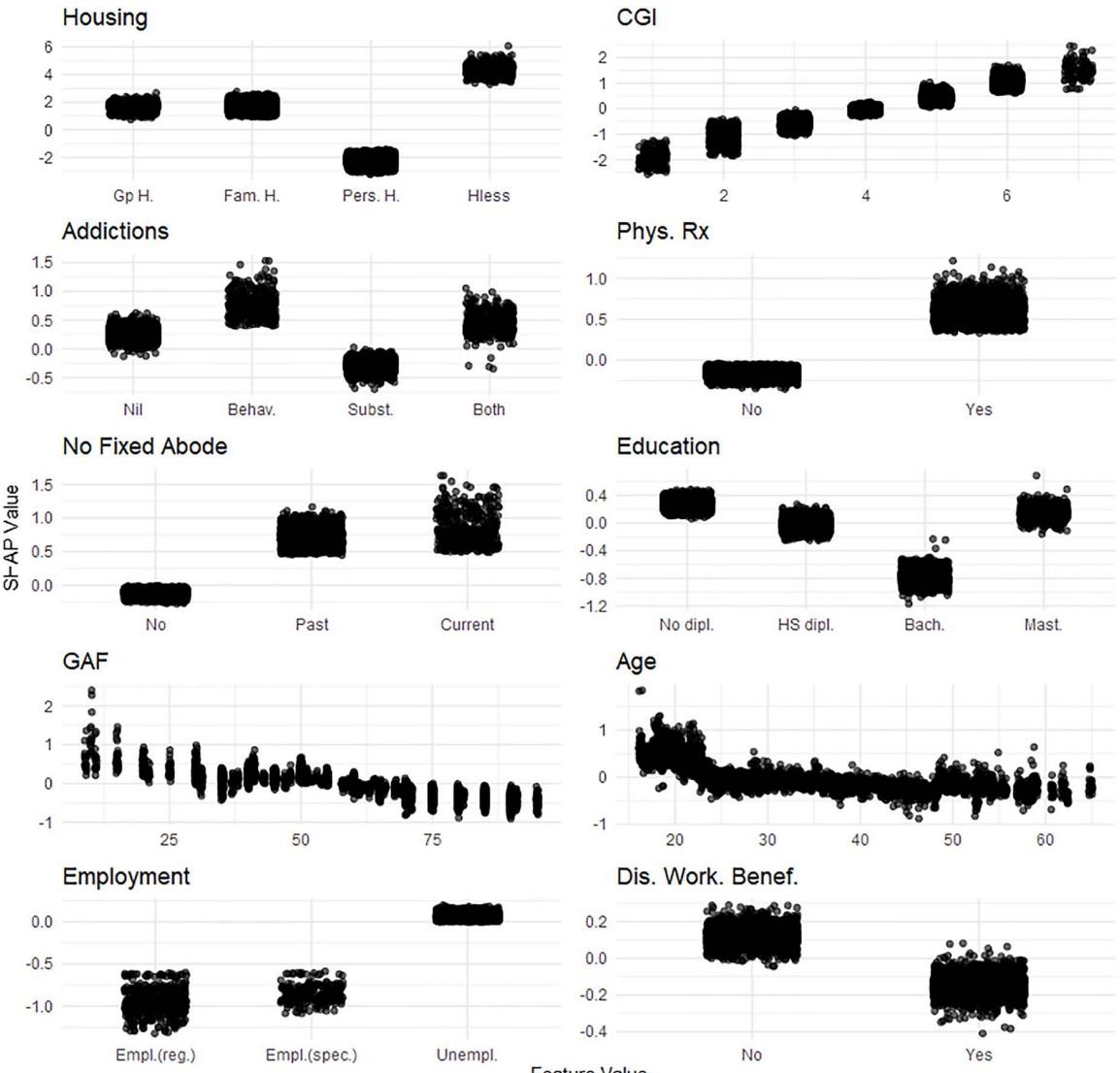

**Fig 1. One-Way SHAP dependence plot of the 10 most influential features for the basic ADL dimension of the Social Autonomy Scale.** Values of the predictor are represented on the x-axis. SHAP values are represented on the y-axis. SHAP values for the 20 imputed datasets are displayed. A higher SHAP value indicates a stronger contribution to increasing the outcome's predicted value. Legend. ADL, activities of daily living; dipl., diploma; HS, High-School; Bach., Bachelor's degree; Mast., Master's degree; rel., relationship; Div./Wid., Divorced/Widowed; Gp H., Group Home; Fam. **H.**, Family Home; Pers. **H.**, Personal Home; Hless, Homeless; Empl., Employed; reg., regular; spec., specialized; Unempl., Unemployed; Dis Work. Benef., Disability Worker Beneficiary; FGA, first generation antipsychotic; SGA, second generation antipsychotic; Dur. Illness, Duration of illness; < 2 yrs, less than 2 years; 10 yrs +, 10 years or more; Psych., Psychiatric; Comorb., Comorbidities; Suicide att., Suicide attempts; Phys., Physical; 4 +, 4 or more; Behav., Behavioral; Subst., Substance; Rx, Treatment; Hx, History; Pub. HC, Public HealthCare; Pr. HC, Private HealthCare; Soc. **W.**, Social Worker; Pat., Patient.

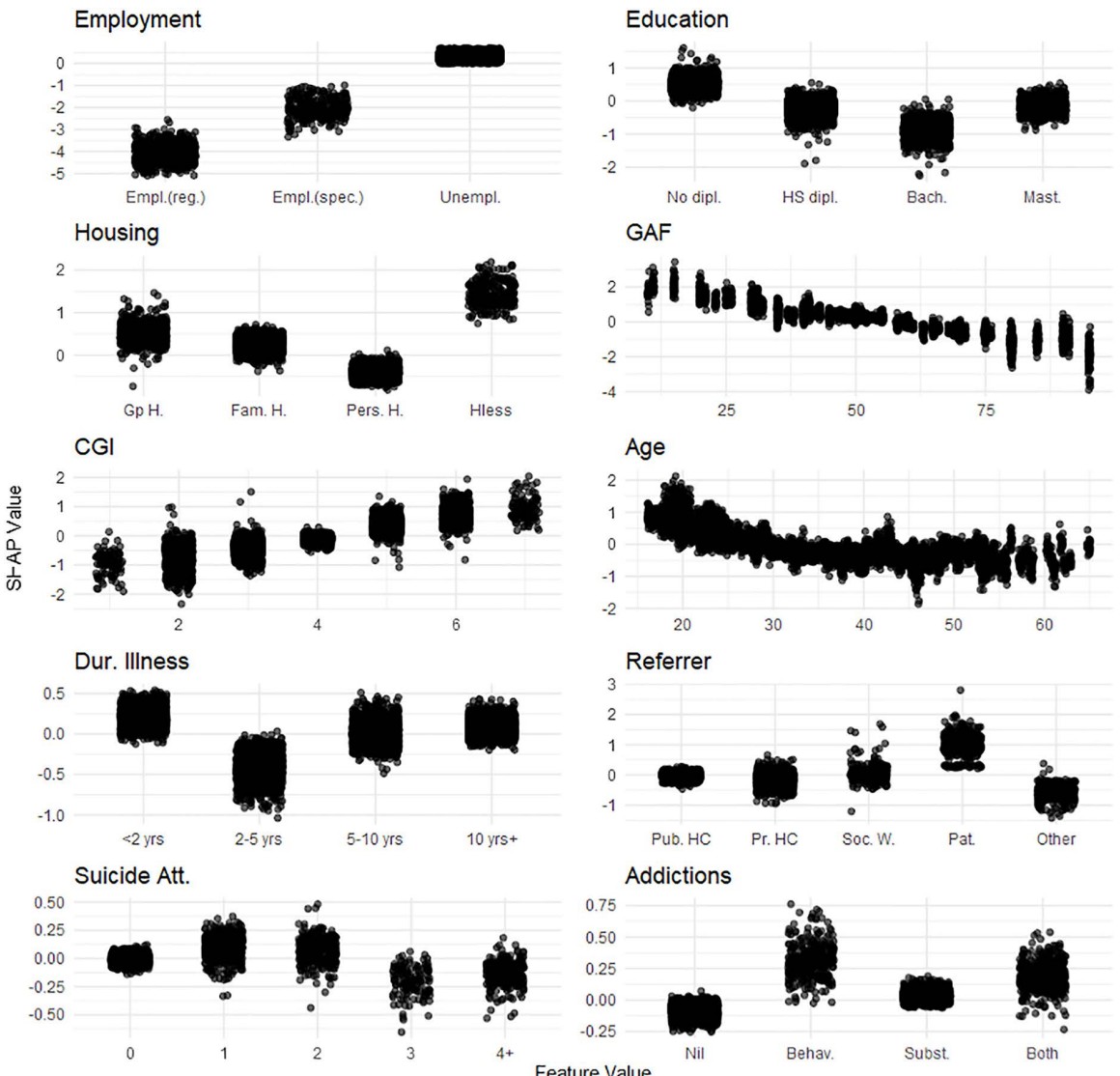

**Fig 2. One-Way SHAP dependence plot of the 10 most influential features for the financial autonomy dimension of the Social Autonomy Scale.** Values of the predictor are represented on the x-axis. SHAP values are represented on the y-axis. SHAP values for the 20 imputed datasets are displayed. A higher SHAP value indicates a stronger contribution to increasing the outcome's predicted value. Legend. dipl., diploma; HS, High-School; Bach., Bachelor's degree; Mast., Master's degree; rel., relationship; Div./Wid., Divorced/Widowed; Gp **H.**, Group Home; Fam. **H.**, Family Home; Pers. **H.**, Personal Home; Hless, Homeless; Empl., Employed; reg., regular; spec., specialized; Unempl., Unemployed; Dis Work. Benef., Disability Worker Beneficiary; FGA, first generation antipsychotic; SGA, second generation antipsychotic; Dur. Illness, Duration of illness; <2 yrs, less than 2 years; 10 yrs +, 10 years or more; Psych., Psychiatric; Comorb., Comorbidities; Suicide att., Suicide attempts; Phys., Physical; 4 +, 4 or more; Behav., Behavioral; Subst., Substance; Rx, Treatment; Hx, History; Pub. HC, Public HealthCare; Pr. HC, Private HealthCare; Soc. **W.**, Social Worker; Pat., Patient.

absence of physical treatment, no history of unstable housing, higher education (excluding the Master's level), higher GAF scores, older age, employment, and being a disabled worker beneficiary (**Fig 1**).

Improved financial autonomy was associated with: being employed (particularly in regular employment), having higher education (though not at the Master's level), stable housing (especially personal housing), higher GAF scores, lower CGI scores, older age, an illness duration of 2–5 years, not self-referring to the service, a greater number of suicide attempts, and the absence of addictions (**Fig 2**).

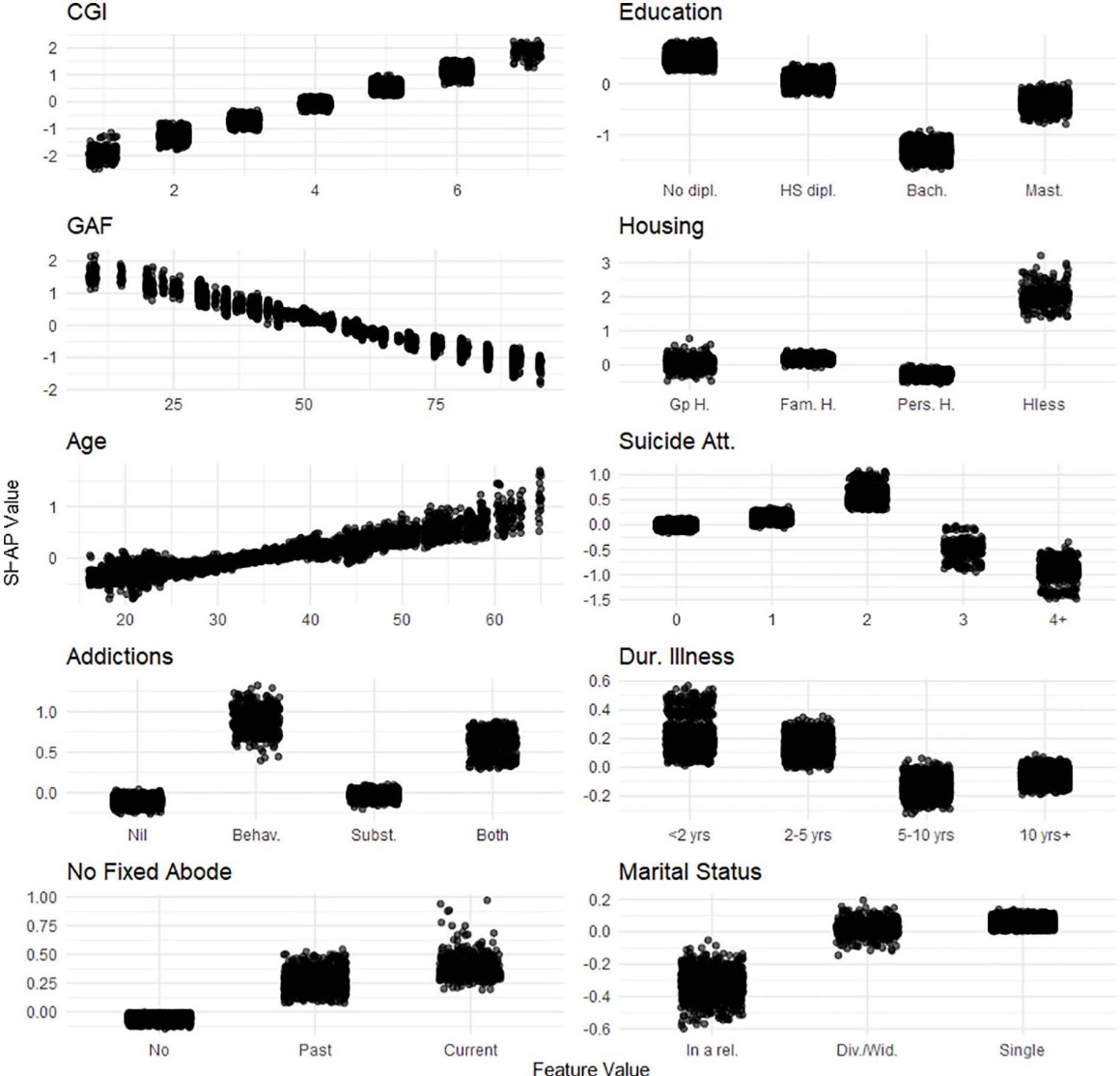

**Fig 3. One-Way SHAP dependence plot of the 10 most influential features for the complex ADL dimension of the Social Autonomy Scale.** Values of the predictor are represented on the x-axis. SHAP values are represented on the y-axis. SHAP values for the 20 imputed datasets are displayed. A higher SHAP value indicates a stronger contribution to increasing the outcome's predicted value. Legend. ADL, activities of daily living; dipl., diploma; HS, High-School; Bach., Bachelor's degree; Mast., Master's degree; rel., relationship; Div./Wid., Divorced/Widowed; Gp H., Group Home; Fam. H., Family Home; Pers. H., Personal Home; Hless, Homeless; Empl., Employed; reg., regular; spec., specialized; Unempl., Unemployed; Dis Work. Benef., Disability Worker Beneficiary; FGA, first generation antipsychotic; SGA, second generation antipsychotic; Dur. Illness, Duration of illness; < 2 yrs, less than 2 years;10 yrs +, 10 years or more; Psych., Psychiatric; Comorb., Comorbidities; Suicide att., Suicide attempts; Phys., Physical; 4 +, 4 or more; Behav., Behavioral; Subst., Substance; Rx, Treatment; Hx, History; Pub. HC, Public HealthCare; Pr. HC, Private HealthCare; Soc. W., Social Worker; Pat., Patient.

For complex ADLs, key predictors included: lower CGI scores, higher education (excluding the Master's level), higher GAF scores, stable housing, younger age, a greater number of suicide attempts, either having a substance addiction or no addictions at all, illness duration of more than five years, and being in a relationship (**Fig 3**).

Finally, factors associated with improved scores on the social and affective relationships dimension included: being in a relationship, lower CGI scores, younger age, personal housing, being on second generation antipsychotic, illness duration

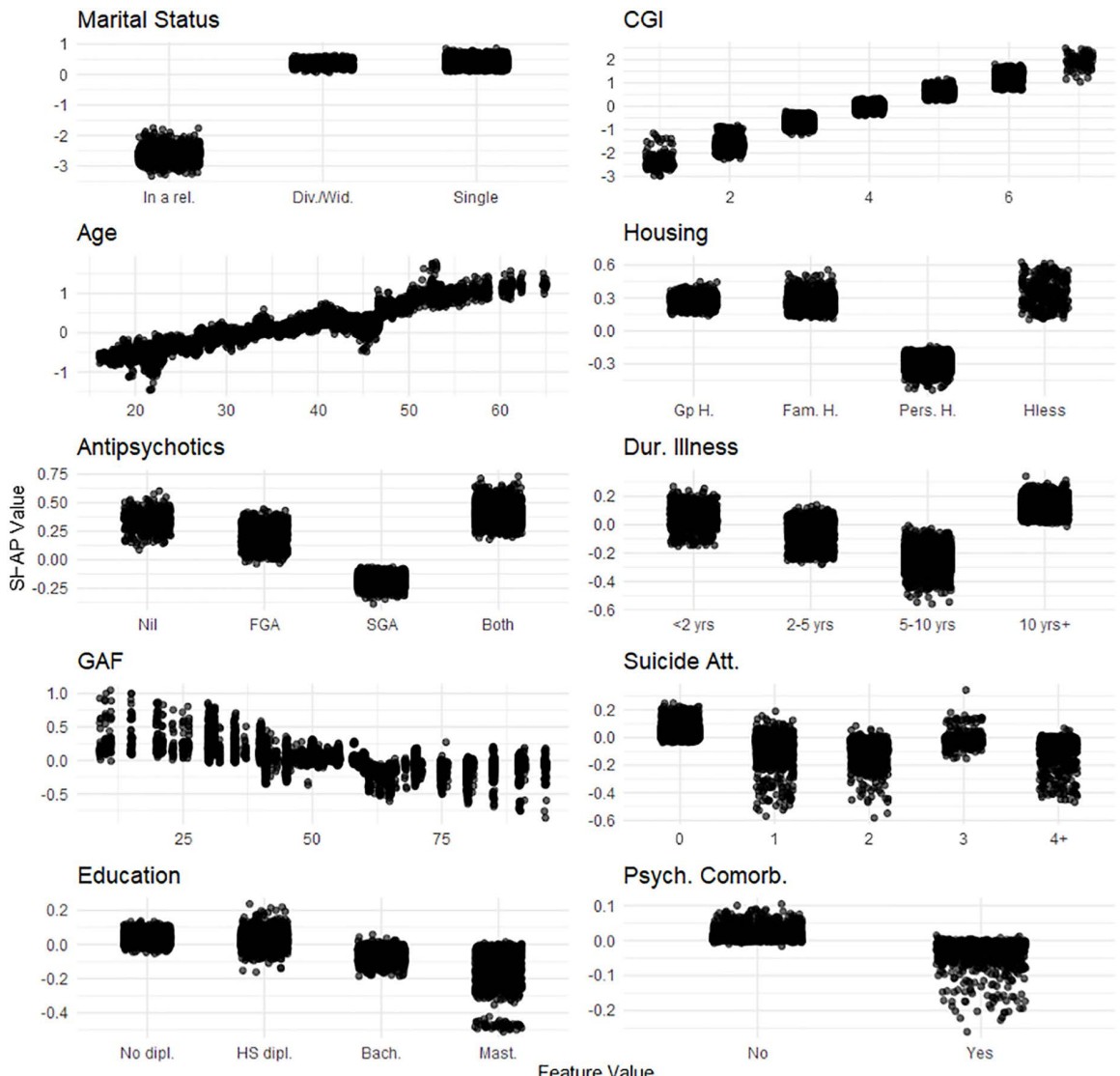

**Fig 4. One-Way SHAP dependence plot of the 10 most influential features for the social and affective relationships dimension of the Social Autonomy Scale.** Values of the predictor are represented on the x-axis. SHAP values are represented on the y-axis. SHAP values for the 20 imputed datasets are displayed. A higher SHAP value indicates a stronger contribution to increasing the outcome's predicted value. Legend. dipl., diploma; HS, High-School; Bach., Bachelor's degree; Mast., Master's degree; rel., relationship; Div./Wid., Divorced/Widowed; Gp **H.**, Group Home; Fam. **H.**, Family Home; Pers. **H.**, Personal Home; Hless, Homeless; Empl., Employed; reg., regular; spec., specialized; Unempl., Unemployed; Dis Work. Benef., Disability Worker Beneficiary; FGA, first generation antipsychotic; SGA, second generation antipsychotic; Dur. Illness, Duration of illness; <2 yrs, less than 2 years;10 yrs +, 10 years or more; Psych., Psychiatric; Comorb., Comorbidities; Suicide att., Suicide attempts; Phys., Physical; 4 +, 4 or more; Behav., Behavioral; Subst., Substance; Rx, Treatment; Hx, History; Pub. HC, Public HealthCare; Pr. HC, Private HealthCare; Soc. **W.**, Social Worker; Pat., Patient.

of 5–10 years, higher GAF scores, having a history of suicide attempts, higher education, and having psychiatric comorbidities (**Fig 4**).

## Discussion

In this study, we investigated whether socio-demographic and basic clinical information routinely collected in clinical practice could be associated with various dimensions of social functioning and activities of daily living (ADLs) in patients with

schizophrenia. Our findings indicate that these factors are moderately informative for basic ADLs (mean R2: 0.35), less effective for financial management 0.14), complex ADLs (0.13) and social/affective relationships (0.16), and not effective for personal care (0.01). Broadly speaking, the most influential socio-demographic and clinical factors across dimensions of functioning include age, marital status, education, housing, employment, clinical site, clinical symptoms, and other clinical characteristics such as antipsychotic treatment, duration of illness, history of suicide attempts, and comorbid addictions – consistent with earlier studies [11].

Compared to previous studies investigating social functioning and ADLs in schizophrenia, the main added values of the current study are: 1) the use of an ensemble machine learning model to gain predictive performance; and 2) the evaluation of model performance on an independent dataset, to provide a robust assessment of predictive validity. Most R2 values indicated relatively modest model performance. As a result, the model has limited utility for identifying individual patients at higher risk of social functioning and ADLs difficulties. Specifically, complex ADLs and domains such as social relationships or financial management may require higher-level cognitive factors, such as executive functions (e.g., inhibition) [26] and social cognition [27,28] – factors that are often not fully assessed in routine clinical practice. In contrast, basic ADLs (such as eating, cleaning and dressing) may be more associated withbasic socio-demographic and clinical variables [29]. Personal care was not well predicted by our model, which may be due to the absence of more nuanced and individualized factors, such as motivation, habits, and environmental supports. Additionally, personal care was the most preserved domain of ADLs in our sample and exhibited the lowest variability, which may have further limited the model's predictive power.

Despite the modest predictive performance of our models, socio-demographic and clinical factors still provide a meaningful, though partial, explanation for the observed variability in functioning abilities. The following briefly reviews socio-demographic and clinical factors influential of social functioning and ADLs. Across the age spectrum, younger individuals typically experience heightened social exploration and stronger social engagement compared to older counterparts [30]. Younger patients may show less difficulty with complex ADL tasks as a result of having less age-related cognitive decline [31]. In contrast, financial autonomy may be easier to establish with advancing age, while having adequate housing faces challenges in accessibility for younger generations [32].

It is unsurprising that being in a relationship is associated with higher scores on the social and complex ADL dimensions. Being in a relationship is a direct indicator of one's capacity for social connection, affective engagement, and interpersonal functioning. In addition, being in a relationship may provide the necessary support and motivation for complex ADLs such as going out or having leisure activities [33].

In addition to providing socioeconomic advantage, education cultivates critical thinking and self-organization, which may contribute to improve functioning [34]. However, compared to reaching a Bachelors degree, a higher education reaching a Masters level was not associated with better ADLs. Individuals with a Masters degree may have higher expectations for their own functioning and independence. If their illness prevents them from utilizing their advanced skills in daily life or employment, this mismatch can lead to heightened frustration, loss of motivation, or even depressive symptoms, which may negatively impact social functioning and ADLs.

Housing, and in particular personal housing, was related to better social functioning and ADLs. A scoping review found that personalized housing residents engage in more domestic tasks and social activities compared to group home residents, directly enhancing social autonomy and ADLs [35]. Comparatively, group homes have been traditionally considered as structured environments that prioritize safety and efficiency over individuality, leading to "institutionalization" (and perhaps stigmatisation) that stifles autonomy [36].

Employment is related to better ADLs in people with schizophrenia, probably because having a job often requires and encourages the use of practical skills – such as time management, personal care, and communication – and provides structure, routine, and motivation, that directly translate to better functioning in daily life. Employment is also associated with improved social functioning, higher self-esteem, and better quality of life [37], all of which may support greater independence in ADLs [38,39].

In terms of clinical factors, lower CGI scores indicate milder symptom severity, which correlates with better capacity for social engagement and ADLs [8]. In the same line, higher GAF scores signify better overall functioning, which encompasses social and occupational performance. Medication-related heterogeneity contributed to predictive performance to a modest extent, though it was more pronounced for social and affective relationships. This is arguably due to clinical efficacy: comparatively to first generation antipsychotics, second generation antipsychotics may alleviate symptoms like emotional dysregulation and cognitive dysfunctions, enabling individuals to engage more effectively in social interactions and ADLs [40]. Mid-range illness duration (vs. shorter or longer) may reflect a stabilization phase, allowing for skill development and coping strategies. Instead, early-stage adaptation may involve greater instability, while chronicity may be associated with accumulated deficits and functional decline. Having addiction comorbidities is linked with increased direct spending on addictive behaviors, debt and legal issues, deteriorated decision-making and self-control, as well as inability to align actions with long-term values, which may explain its association with poorer financial autonomy.

We also observed unexpected relationships between a set of clinical factors and various dimensions of functioning. Individuals with a greater number of suicide attempts, those exhibiting psychiatric comorbidities or substance addiction demonstrated *enhanced* social functioning and/or ADL abilities. This may be due to a heightened access to support systems in individuals with more severe or complex disorders. This phenomenon, which can be interpreted through the lens of service engagement bias, suggests that patients presenting with more severe or high-risk clinical profiles are more frequently prioritized for intensive multidisciplinary care, vocational rehabilitation, and structured community support. While markers such as multiple suicide attempts or comorbid substance use indicate a profound clinical burden, they also serve as significant drivers for clinical entry into protected care pathways. Consequently, these individuals may receive a higher level of medical and social intervention compared to patients with milder symptomatic profiles, who may be less likely to trigger such rigorous support protocols and thus risk falling through the gaps of the care system. Our findings that individuals referred by professionals (compared to self-referrers) and beneficiaries of disability worker assistance exhibit better functioning further support this interpretation.

Based on these findings, some public health recommendations can be made. First, we suggest interventions targeting clinical symptoms, improving housing conditions, particularly by promoting access to personal housing, supporting employment and/or assistance in obtaining disability benefits. Second, as some non-modifiable factors are associated with lower social functioning and difficulties in ADLs, specialized care should be offered to high-need patients, such as those with lower levels of education. Third, additional strategies may be necessary to address factors with paradoxical associations. For example, individuals with a high level of education, shorter illness duration, no history of suicide attempts, and no psychiatric or addiction comorbidities may still require a significant level of support to enhance their sense of hope, self-esteem, insight into their disorders, and access to appropriate services.

## Strengths and limitations

Strengths of the study include its large sample size and number of predictors, as well as its optimal methodology, including the use of a large library of base learners and assessment of performance on independent datasets.

However, our study is not without limitations. First, our measures of social functioning and ADLs were based on clinician rather than patient assessments. Consequently, our findings may be susceptible to observer bias. There is evidence however, that patients may underestimate their own disabilities [41], and that subjective assessments tend to show consistent correlations with mood, rather than with objective life situations [42]. Second, although our set of sociodemographic and basic clinical predictors was extensive, it may still have omitted certain important factors. Notably, we did not include measures of negative symptoms, medication side effects, specific types of physical issues, or cognitive performance [43]. Additional factors such as insight into one's disorder [44], psychiatric medications other than antipsychotics, and medication adherence may have also influenced functioning. We acknowledge that some of the variables routinely collected were not available in our database, and might have improved the predictive ability of our models. The

omission of these clinical drivers not only reduces the model's discriminatory power but also restricts the interpretation of our results. Specifically, our variable importance metrics (SHAP values) may be subject to residual confounding, which potentially masks the distinct contributions of our predictors to patient functioning. Third, the social autonomy scale we used is primarily utilized and validated in France, which may limit the generalizability of our findings. While the original validation dates back to 1998, we are currently re-evaluating the instrument's psychometric properties using contemporary datasets and advanced analytical techniques. Fourth, our analysis included only patients with complete data on social functioning and ADLs, which may have introduced selection bias. This is particularly relevant given that non-utilization of services is significantly associated with functional impairment and willingness to engage in psychosocial rehabilitation [45]. Fifth, some relationships between variables may appear counterintuitive, such as the association between substance use disorders and better basic ADLs. Additionally, certain variables may be linked to social functioning and ADLs due to reverse causation, where the outcome may influence the predictor rather than the other way around. It is important to emphasize that our study was not designed to establish causality; rather, our aim was to assess the predictive value of variables commonly collected in clinical practice. In particular, we emphasize that SHAP values reflect our predictive model behavior rather than the inherent causal significance of the predictors. Sixth, performance metrics are derived exclusively from the testing set, which yielded lower predictive performance than the training set. This discrepancy provides clear evidence of overfitting, suggesting that the model captured idiosyncratic noise specific to the training data rather than exclusively learning generalizable clinical associations. In a clinical context, relying on training performance would be highly misleading, as it introduces an optimistic bias that overestimates the model's utility. Therefore, we emphasize testing set performance as the only trustworthy indicator of real-world applicability; despite the modest R2 values, this testing performance represents a realistic appraisal of how the model will perform when presented with new, unseen patients. Seventh, it is important to acknowledge that the clinical heterogeneity observed in our study may partially stem from underlying variability in neurobiological processes. While our model is based on socio-demographic and clinical characteristics, it does not incorporate neural dynamics or brain imaging data, as these measures fall outside the scope of our behavioral and clinical focus. Future interdisciplinary investigations would be a logical next step to determine whether these behavioral groupings correspond to distinct neurobiological phenotypes.

## Conclusion

Our findings indicate an association between socio-demographic and standard clinical variables routinely assessed in practice and outcomes in social functioning and ADLs. However, these variables account for only a limited proportion of the observed variance.. This underscores the need for more specialized and precise assessments, particularly those that evaluate negative symptoms and cognitive functioning, to better understand and address patient functioning. Based on our findings, we recommend targeted interventions focused on improving clinical symptoms, housing conditions, especially by facilitating access to personal housing, and supporting employment or, where necessary, assisting with disability benefits. Finally, clinicians should remain vigilant and not assume that patients with seemingly protective factors, such as higher education, shorter illness duration, or absence of comorbidities, do not require further support; these individuals may still benefit from efforts to enhance their access to appropriate services in order to improve their social functioning and ADLs.

## Supporting information

**S1 Table. Performance of each basis learner and of the ensemble algorithm.**
(DOCX)

**S2 Table. Contribution of each base learner to the SuperLearner ensemble.**
(DOCX)

**S1 Fig. Mean absolute SHAP value for the four dimensions of the Social Autonomy Scale showing meaningful performance of our model.**

(DOCX)

**Ethics statement**

## Author contributions

**Conceptualization:** Guillaume Barbalat.

**Data curation:** Julien Plasse, Isabelle Chéreau-Boudet, Benjamin Gouache, Nathalie Guillard-Bouhet, Emilie Legros-Lafarge.

**Formal analysis:** Guillaume Barbalat.

**Investigation:** Guillaume Barbalat, Emilie Legros-Lafarge.

**Methodology:** Guillaume Barbalat, Julien Plasse.

**Project administration:** Guillaume Barbalat, Emilie Legros-Lafarge, Nicolas Franck.

**Resources:** Julien Plasse, Nicolas Franck.

**Software:** Guillaume Barbalat, Julien Plasse.

**Supervision:** Nicolas Franck.

**Validation:** Nicolas Franck.

**Visualization:** Guillaume Barbalat, Nicolas Franck.

**Writing – original draft:** Guillaume Barbalat.

**Writing – review & editing:** Guillaume Barbalat, Julien Plasse, Nicolas Franck.

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
