## [Decision Letter · Decision Letter 0]

19 Feb 2026

PONE-D-25-50813Can routinely collected clinical and sociodemographic characteristics predict social functioning and activities of daily living in schizophrenia? A machine learning approach.PLOS One?

Dear Dr. Barbalat,

Thank you for submitting your manuscript to PLOS ONE. After careful consideration, we feel that it has merit but does not fully meet PLOS ONE’s publication criteria as it currently stands. Therefore, we invite you to submit a revised version of the manuscript that addresses the points raised during the review process.

The authors are requested to undertake **major revisions** in accordance with the reviewers’ comments.in accordance with the reviewers’ comments.in accordance with the reviewers’ comments.in accordance with the reviewers’ comments.

A letter that responds to each point raised by the academic editor and reviewer(s). You should upload this letter as a separate file labeled ’Response to Reviewers’.A marked-up copy of your manuscript that highlights changes made to the original version. You should upload this as a separate file labeled ’Revised Manuscript with Track Changes’.An unmarked version of your revised paper without tracked changes. You should upload this as a separate file labeled ’Manuscript’.

We look forward to receiving your revised manuscript.

Kind regards,

De-Chih Lee, Ph.D.

Academic Editor

PLOS One

Journal Requirements:

1. Please ensure that your manuscript meets PLOS ONE’s style requirements, including those for file naming. The PLOS ONE style templates can be found at

Additional Editor Comments:

The authors are requested to undertake major revisions in accordance with the reviewers’ comments.

Reviewers’ comments:

Reviewer’s Responses to Questions

**Comments to the Author**

1. Is the manuscript technically sound, and do the data support the conclusions?

Reviewer #1: Yes

Reviewer #2: Partly

Reviewer #3: Yes

Reviewer #4: Partly

Reviewer #5: Partly

2. Has the statistical analysis been performed appropriately and rigorously?

Reviewer #1: N/A

Reviewer #2: No

Reviewer #3: Yes

Reviewer #4: Yes

Reviewer #5: Yes

3. Have the authors made all data underlying the findings in their manuscript fully available?

Reviewer #1: Yes

Reviewer #2: No

Reviewer #3: No

Reviewer #4: Yes

Reviewer #5: Yes

4. Is the manuscript presented in an intelligible fashion and written in standard English?

Reviewer #1: Yes

Reviewer #2: Yes

Reviewer #3: Yes

Reviewer #4: Yes

Reviewer #5: Yes

Reviewer #1: This manuscript addresses an important topic: whether routinely collected clinical and sociodemographic characteristics can predict social functioning and activities of daily living in schizophrenia using a machine learning approach.

The manuscript is clearly written, and the authors benefit from a large sample size and the use of advanced machine learning techniques. However, several issues require clarification before the manuscript can be considered for publication.

• Many variables known to predict functioning in schizophrenia (e.g., negative symptoms, cognitive performance, medication side effects) were not available. The authors should discuss how the absence of these variables may have affected model performance and interpretation.

• The manuscript should clarify how the sample size was determined

• Clarify how categorical variables were encoded.

• Specify whether the train/test split was stratified by center.

• Some results require deeper discussion, such as:

• More suicide attempts being associated with better functioning.

• Psychiatric comorbidities being associated with better social relationships.

Reviewer #2: The study focuses on schizophrenia; however, the current manuscript considers only behavioral and demographic measures, and the Introduction is insufficiently developed given the complexity of the disorder.

Schizophrenia is a heterogeneous and stage-dependent disorder, with well-documented variability across disease phases and medication status. Contemporary neuroscience research increasingly aims to model the underlying neural mechanism disruptions and to translate them into quantitative biomarkers as following study: https://doi.org/10.5455/apd.205512,

In this context, the authors are encouraged to substantially expand the Introduction by discussing the full range of variables that characterize schizophrenia, with particular emphasis on studies investigating its neural signatures. It would also be appropriate for the authors to explicitly acknowledge that the current work adopts a limited perspective, focusing primarily on behavioral and demographic dimensions.

In the Participants section, age-related details should be reported more rigorously. For example, the authors should provide mean age and standard deviation separately for women and men, rather than reporting aggregated statistics. In fact, in psychiatric disorders, medication status can substantially alter intrinsic brain dynamics, which in turn is expected to influence social behavior and related phenotypes as stated in following study: https://dusunenadamdergisi.org/storage/upload/pdfs/1585638731-en.pdf

Consequently, heterogeneity in medication use among the modeled patient cohort may have a non-trivial impact on the reported results. The authors are therefore encouraged to systematically examine the dataset with respect to medication-related heterogeneity and to discuss its potential effects on both the predictive modeling outcomes and their interpretation.

The manuscript should clearly describe the clinical heterogeneity of the patient cohort. Relevant information includes: duration of illness (e.g., number of years since schizophrenia diagnosis), medication status (e.g., long-term medication use, newly initiated treatment, different medication classes), presence of comorbid conditions.

Explicitly reporting these factors is essential for interpreting the results and assessing their generalizability.

The predictive models employed in the study are not sufficiently explained. The authors should clarify whether user-chosen or optimized parameters were used, whether patients were subgrouped or categorized, and how many participants were included in each group or category. In addition, it should be stated how many features were used to represent each individual.

Given that the model appears to assume similarity among participants sharing certain characteristics, an important question remains unanswered: did the authors examine whether these individuals also exhibit similar neural brain dynamics or patterns? If not, do the authors plan to address this issue in a supportive and confirmatory future work framework?

Please specify the software platform(s) on which the predictive models were implemented.

Finally, the manuscript does not address the stability and consistency of the proposed model. Were reliability analyses such as test–retest evaluations or intra-class correlation coefficients (ICC) conducted? If not, the authors should discuss this limitation. Without quantitative evidence of stability, it is difficult to assess the robustness of the proposed approach.

Reviewer #3: This manuscript presents a methodologically sound and ethically robust analysis of whether routinely collected clinical and sociodemographic variables can predict social functioning and activities of daily living in schizophrenia. Using a large multicenter dataset and appropriate machine-learning methods with independent validation, the authors provide a transparent and balanced assessment of predictive performance without overstating clinical utility. Limitations and counterintuitive findings are carefully acknowledged and interpreted cautiously, minimizing the risk of biased or misleading conclusions. Overall, the study makes a credible and incremental contribution.

** The data is not publicly accessible as the author mention about public data protection act, however the authors states that upon request it will be available.

Reviewer #4: The study addresses a clinically relevant question using a sophisticated machine learning pipeline and a well-characterized multicentric dataset. The manuscript is clearly written and methodologically ambitious. However, we have identified three areas-distinct from the limitations already acknowledged-that would substantially strengthen the scientific contribution and clinical applicability of this work.

(i) Inclusion of a Psychiatric or Normative Comparator Group

The current design describes predictors of functioning within schizophrenia, but cannot determine whether these predictors are disorder-specific or transdiagnostic. We recommend that the authors either: (a) include a control group of individuals with other severe mental illnesses (e.g., bipolar disorder, major depressive disorder) from the REHABase or a linked dataset, or (b) explicitly reframe the study scope to reflect that findings are descriptive of a schizophrenia cohort only. If the latter, this should be clearly stated in the title and abstract to avoid overgeneralization.

(ii) Temporal Validation to Support Prognostic Claims

The manuscript implies prognostic utility, yet the analysis is cross-sectional. We strongly encourage the authors to explore whether any follow-up data exist within REHABase (e.g., SAS reassessments at 6 or 12 months). Even a preliminary analysis of change scores or functional deterioration/progression would significantly enhance the paper’s translational value. If such data are not available, we recommend tempering claims about “predicting” outcomes in favor of “associating” or “explaining variance in.”

(iii) Sensitivity and Stability Analysis for SHAP Interpretations

SHAP is a powerful interpretability tool, but its reliability is contingent on model and data stability. Given the variability in base learner weights and the risk of overfitting, we request that the authors report a measure of SHAP value stability (e.g., rank correlation of feature importance across imputed datasets, or 95% bootstrap intervals for mean absolute SHAP). This would allow readers to assess the confidence with which specific predictors (e.g., Master’s degree paradox can be translated into clinical heuristics.

Reviewer #5: This manuscript investigates whether routinely collected clinical and sociodemographic variables can predict social functioning and activities of daily living (ADLs) in individuals with schizophrenia using a SuperLearner machine-learning framework. The topic is clinically relevant and aligns with current efforts to integrate data-driven approaches into psychosocial rehabilitation research. The use of a large multicenter dataset (n=948) and ensemble modeling represents a notable methodological strength. Overall, the manuscript is clearly written and methodologically structured. However, several conceptual, methodological, and reporting issues require clarification before the manuscript can be considered for publication. In particular, concerns relate to model interpretation, generalizability, data availability compliance, and the clinical meaning of predictive performance.

I provide detailed comments below.

1. Prediction vs. causation: The manuscript uses a predictive framework, but parts of the Discussion imply causal relationships. Please revise interpretations to avoid causal language.

2. Model performance and overfitting: Test-set performance appears modest with a noticeable gap between training and testing results. Provide clearer discussion on overfitting and the clinical relevance of the predictive accuracy.

3. Methodological transparency: Please clarify feature selection procedures, cross-validation design, and whether hyperparameter tuning was nested to avoid data leakage.

4. Missing data handling: Specify the imputation method, variables included in the imputation model, and how estimates were combined across imputed datasets.

5. Interpretation of SHAP results: Emphasize that SHAP values reflect model behavior rather than causal importance of predictors.

6. Data availability statement: Further clarification is needed to ensure compliance with PLOS ONE’s data sharing policy (e.g., controlled-access repository or clear access procedure).

7. Generalizability: Expand discussion on limitations related to the Social Autonomy Scale and potential clinician-rating bias.

.

Reviewer #1: No

Reviewer #2: No

Reviewer #3: No

Reviewer #4: **Yes:** Dr Saleem IqbalDr Saleem IqbalDr Saleem IqbalDr Saleem Iqbal

Reviewer #5: No

---

## [Author Response · Author response to Decision Letter 1]

12 Mar 2026

Please see the response to reviewers file.

---

## [Decision Letter · Decision Letter 1]

1 Apr 2026

Are routinely collected clinical and sociodemographic characteristics associated with social functioning and activities of daily living in schizophrenia? A machine learning approach descriptive of a schizophrenia cohort.

PONE-D-25-50813R1

Dear Dr. Barbalat,

We’re pleased to inform you that your manuscript has been judged scientifically suitable for publication and will be formally accepted for publication once it meets all outstanding technical requirements.

An invoice will be generated when your article is formally accepted. Please note, if your institution has a publishing partnership with PLOS and your article meets the relevant criteria, all or part of your publication costs will be covered. Please make sure your user information is up-to-date by logging into Editorial Manager at Editorial Manager® and clicking the ‘Update My Information’ link at the top of the page. For questions related to billing, please contact  and clicking the ‘Update My Information’ link at the top of the page. For questions related to billing, please contact  and clicking the ‘Update My Information’ link at the top of the page. For questions related to billing, please contact  and clicking the ‘Update My Information’ link at the top of the page. For questions related to billing, please contact billing support....

Kind regards,

De-Chih Lee, Ph.D.

Academic Editor

PLOS One

Additional Editor Comments (optional):

Reviewers’ comments:

Reviewer’s Responses to Questions

**Comments to the Author**

Reviewer #1: All comments have been addressed

Reviewer #4: All comments have been addressed

Reviewer #5: All comments have been addressed

2. Is the manuscript technically sound, and do the data support the conclusions?

Reviewer #1: Yes

Reviewer #4: Yes

Reviewer #5: Yes

3. Has the statistical analysis been performed appropriately and rigorously?

Reviewer #1: Yes

Reviewer #4: Yes

Reviewer #5: Yes

4. Have the authors made all data underlying the findings in their manuscript fully available?

Reviewer #1: Yes

Reviewer #4: Yes

Reviewer #5: Yes

5. Is the manuscript presented in an intelligible fashion and written in standard English?

Reviewer #1: Yes

Reviewer #4: Yes

Reviewer #5: Yes

Reviewer #1: Thank you for your efforts in revising the manuscript. I have reviewed the revised version and found that the manuscript has been significantly improved in terms of clarity, structure, and overall quality.

Reviewer #4: The authors have addressed all major and minor concerns raised in the previous review round in a clear and satisfactory manner. The study presents a well-conducted and methodologically sound application of machine learning to investigate determinants of social functioning and activities of daily living in schizophrenia, using routinely collected clinical and sociodemographic data. The manuscript can be accepted in its current form.

Reviewer #5: The authors have adequately addressed all my comments and significantly improved the clarity, methodological transparency, and interpretation of the manuscript. I have no further concerns and recommend acceptance.

.

Reviewer #1: No

Reviewer #4: **Yes:** Prof. Saleem IqbalProf. Saleem IqbalProf. Saleem IqbalProf. Saleem Iqbal

Reviewer #5: No

---

## [Editor Report · Acceptance letter]

PONE-D-25-50813R1

PLOS One

Dear Dr. Barbalat,

I’m pleased to inform you that your manuscript has been deemed suitable for publication in PLOS One. Congratulations! Your manuscript is now being handed over to our production team.

Lastly, if your institution or institutions have a press office, please let them know about your upcoming paper now to help maximize its impact. If they’ll be preparing press materials, please inform our press team within the next 48 hours. Your manuscript will remain under strict press embargo until 2 pm Eastern Time on the date of publication. For more information, please contact onepress@plos.org.

Kind regards,

on behalf of

Dr. De-Chih Lee

Academic Editor

PLOS One